

# Enrichment in conservative amino acid changes among fixed and standing missense variations in slowly evolving proteins

Mingrui Wang[1], Dapeng Wang[2,3], Jun Yu[2] and Shi Huang[1]

[1] Center for Medical Genetics & Hunan Key Laboratory of Medical Genetics, School of Life Sciences, Central South University, Changsha, Hunan, P.R. China
[2] CAS Key Laboratory of Genome Sciences and Information, Beijing Institute of Genomics, Chinese Academy of Sciences, Beijing, Beijing, P.R. China
[3] Current affiliation: LeedsOmics, University of Leeds, Leeds, UK

Corresponding author
Shi Huang, huangshi@sklmg.edu.cn

## ABSTRACT

The process of molecular evolution has many elements that are not yet fully understood. Evolutionary rates are known to vary among protein coding and noncoding DNAs, and most of the observed changes in amino acid or nucleotide sequences are assumed to be non-adaptive by the neutral theory of molecular evolution. However, it remains unclear whether fixed and standing missense changes in slowly evolving proteins are more or less neutral compared to those in fast evolving genes. Here, based on the evolutionary rates as inferred from identity scores between orthologs in human and Rhesus Macaques (*Macaca mulatta*), we found that the fraction of conservative substitutions between species was significantly higher in their slowly evolving proteins. Similar results were obtained by using four different methods of scoring conservative substitutions, including three that remove the impact of substitution probability, where conservative changes require fewer mutations. We also examined the single nucleotide polymorphisms (SNPs) by using the 1000 Genomes Project data and found that missense SNPs in slowly evolving proteins also had a higher fraction of conservative changes, especially for common SNPs, consistent with more non-conservative substitutions and hence stronger natural selection for SNPs, particularly rare ones, in fast evolving proteins. These results suggest that fixed and standing missense variants in slowly evolving proteins are more likely to be neutral.

## INTRODUCTION

Since the early 1960s, protein sequence comparisons have become increasingly important in molecular evolutionary research (*Doolittle & Blombaeck, 1964*; *Fitch & Margoliash, 1967*; *Margoliash, 1963*; *Zuckerkandl & Pauling, 1962*). An apparent relationship between protein sequence divergence and time of separation led to the molecular clock hypothesis, which assumes a constant and similar evolutionary rate among species (*Kumar, 2005*; *Margoliash, 1963*; *Zuckerkandl & Pauling, 1962*). Thus, sequence divergence between species is thought

to be largely a function of time. The molecular clock, in turn, led Kimura to propose the neutral theory to explain nature: sequence differences between species were thought to be largely due to neutral changes rather than adaptive evolution (*Kimura, 1968*). However, the notion of a molecular clock may be unrealistic since it predicts a constant substitution rate as measured in generations, whereas the observed molecular clock is measured in years (*Ayala, 1999*; *Pulquerio & Nichols, 2007*). The neutral theory remains an incomplete explanatory theory (*Hu et al., 2013*; *Kern & Hahn, 2018*).

Evolutionary rates are known to vary among protein coding and non-coding DNAs. The neutral theory posits that the substitution rate under selective neutrality is expected to be equal to the mutation rate (*Kimura, 1983*). If mutations/substitutions are not neutral or are under natural selection, the substitution rate would be affected by the population size and the selection coefficient, which are unlikely to be constant among all lineages. Slowly evolving genes are well known to be under stronger purifying or negative selection as measured by using dN/dS ratio, which means that a new mutation has a lower probability of being fixed (*Cai & Petrov, 2010*). However, negative selection as detected by the dN/dS method is largely concerned with non-observed mutations and says little about the fixed or observed variations, and most molecular evolutionary approaches such as phylogenetic and demographic inferences are concerned with observed variants. It remains to be determined whether fixed and standing missense substitutions in slowly evolving genes are more or less neutral relative to those in fast evolving genes.

We here examined the fraction of conservative substitutions (amino acid replacement in a protein that changes a given amino acid to a different amino acid with similar biochemical properties) in proteins of different evolutionary rates. We compared the protein orthologs of two relatively closely related species, *Homo sapiens* and *Macaca mulatta,* to obtain values of percentage identity to represent evolutionary rates. We found that the proportion of conservative substitutions between species was higher in the slowest evolving set of proteins than in faster evolving proteins. Using datasets from the 1000 Genomes (1KG) Project Phase 3 dataset (*Auton et al., 2015*), we also found that missense single nucleotide polymorphisms (SNPs) from the slowest evolving set of proteins, especially those with high minor allele frequency (MAF), were enriched with conservative amino acid changes, consistent with these changes being under weaker natural selection.

## MATERIAL AND METHODS

### Classification of proteins as slowly and fast evolving

The identification of slowly evolving proteins and their associated SNPs was done as previously described (*Yuan et al., 2017*). Briefly, we collected the whole genome protein data of *Homo sapiens* (version 36.3) and *Macaca mulatta* (version 1) from the NCBI FTP site, and then compared the human protein to the monkey protein using local BLASTP program at a cutoff of 1E-10. We only retained one human protein with multiple isoforms, and chose the monkey protein with the most significant *E*-value as the orthologous counterpart of each human protein. The aligned proteins were ranked by percentage identities. Proteins that show the highest identity between human and monkey were

included in the set of slowly evolving (including 423 genes >304 amino acid in length with 100% identity and 178 genes >1,102 amino acid in length with 99% identity between monkey and human). The rest are all considered fast evolving proteins. The cutoff criterion was based on the empirical observation of low substitution saturation, and the finding that missense SNPs from the slow set of proteins produced genetic diversity patterns that were distinct from those found in the fast set (*Yuan et al., 2017*). The BLASTP alignment program is not expected to produce very different results from other programs, especially for highly conserved proteins. We have limited our analysis to high identity orthologs with length >200 amino acid and percent identity >60% between monkey and human. So, variation in alignment is not expected to affect comparing our analysis to others.

## SNP selection

We downloaded the 1KG phase 3 data and assigned SNP categories using ANNOVAR (*Auton et al., 2015*). We then picked out the missense SNPs located in the slow evolving set of genes from the downloaded VCF files (*Yuan et al., 2017*). MAF was derived from AF (alternative allele frequency) values from the VCF files. Missense SNPs in fast evolving genes included all those from 1KG that are not from the slowly evolving set.

## Scoring conservative amino acid replacements

We downloaded the Consensus CDS data of *Homo sapiens* (version 36.3, https://www.ncbi.nlm.nih.gov/projects/CCDS/CcdsBrowse.cgi) and CDS data of *Macaca mulatta* (version 1.0, https://jul2016.archive.ensembl.org/Macaca_mulatta/Info/Index) and searched for related protein pairs using BLASTP program in BLAST+ 2.10.0. Human proteins were used as queries to search in the monkey protein database. We recorded the IDs of query and the first hit (ranked by *E*-values), query length, alignment span, the number of matching amino acids, the combined number of gaps in the query and hit proteins, and the number and type of amino acid substitutions.

For fixed substitutions as revealed by BLASTP, conservative changes were scored by using four different matrixes. The BLOSUM62 matrix has a scoring range from −3 to 3 (−3, −2, −1, 0, 1, 2, 3) with higher positive values representing more conservative changes (*Pearson, 2013*). We assigned each amino acid mutation a score and we used score >0 to denote conservative changes in cases where the number of conservative changes is enumerated. As the BLOSUM62 matrix does not take into account the effect of substitution probability (the fact that conservative changes require fewer mutations), we also used three other matrixes to score conservative amino acid replacements that have removed the impact of substitution probability, including the "EX" matrix (*Yampolsky & Stoltzfus, 2005*), which is based on laboratory mutagenesis, and the two physicochemical matrices in *Braun (2018)* and *Pandey & Braun (2020)*: delta_V (normalized change in amino acid side chain volume), and delta_P (normalized change in amino acid side chain polarity) (*Braun, 2018*; *Pandey & Braun, 2020*). All three matrixes in spreadsheets are available from GITHUB (https://github.com/ebraun68/clade_specific_prot_models). Specifically, the EX matrix (or, more accurately, a normalized symmetric version of the EX matrix) is in the excel spreadsheet "EX_matrix_sym.xlsx"; the delta_V and delta_P matrices can be found in

one of the sheets (the sheet called "Exchanges") in the file "exchange_Pandey_Braun.xlsx". All three of the matrixes are normalized to range from zero to one. To be comparable to the BLOSUM62 matrix, we generated integer versions of these three matrixes by multiplying by 10, subtracting 5, and then rounding to the nearest integer. Here the matrix values range from $-5$ to $+5$ with higher positive values representing more conservative changes. For EX matrix, we used score $>2$ to denote conservative changes. For delta_V and delta_P matrixes, we used score $>3$ to denote conservative changes. In this way of using different cutoff scores to represent conservative changes, we could keep the fraction of conservative changes close to 0.5 for each of the four different matrixes.

## Statistics

Chi-squared test was performed using GraphPad Prism 6.

# RESULTS

## Fixed amino acid substitutions and evolutionary rates of proteins

We determined the evolutionary rates of proteins in the human genome by the percentage of identities between human proteins and their orthologs in *Macaca mulatta* as described previously (*Yuan et al., 2017*). We then divided the proteins into several groups of different evolutionary rates, and compared the proportion of conservative amino acid substitutions in each group.

The mismatches between two species would have one of the two residues or alleles as ancestral, in the case of slowly evolving proteins yet to reach mutation saturation (no independent mutations occurring at the same site among species and across time), and so a mismatch due to conservative changes would involve a conservative mutation during evolution from the ancestor to extant species. But at mutation saturation for fast evolving proteins, where a site had encountered multiple mutations across taxa and time, while a drastic substitution would necessarily involve a non-conservative mutation, it is possible for a conservative substitution to result from at least two independent non-conservative mutations (if the common ancestor has Arg at some site, a drastic mutation event at this site occurring in each of the two species, Arg to Leu in one and Arg to Ile in the other, may lead to a conservative substitution of Leu and Ile). Thus, a conservative substitution at mutation saturation just means less physical and chemical differences between the two species concerned and says little about the actual mutation events. A lower fraction of conservative substitutions at saturation for fast evolving proteins would mean more physical and chemical differences between the two species, which may more easily translate into functional differences for natural selection to act upon.

To verify that the slowest evolving proteins with length $>1,102$ amino acids and percentage identity $>99\%$ are distinct from the fast set, we first compared proteins with length $>1,102$ amino acids with no gaps in alignment (Table 1, Fig. 1A) or with gaps (Table 1, Fig. 1B) divided into four groups of different percentage identity between human and monkey, $>99\%$, $98$–$99\%$, $96$–$98\%$, and $87$–$97\%$. We used four different scoring matrixes to give each amino acid change a rank score in terms of how conservative the change is, BLOSUM62 (*Pearson, 2013*), EX (*Yampolsky & Stoltzfus, 2005*), delta_V, and

**Table 1 Relationship between evolutionary rates and the conservative nature of fixed amino acid substitutions.** Evolutionary rates of proteins in the human genome are represented by the percentage of identities between human proteins and their orthologs in *Macaca* monkey. The proteins are divided into groups of different evolutionary rates, and the proportion of conservative amino acid mismatches in each group are shown for the four different ranking matrixes. Not all proteins encoded by the macaque and human genomes are considered because some proteins do not have easily identifiable orthologs. Also, we limited our analysis to proteins that have length >200 amino acids and show >60% identity between macaque and human in order to reduce the chance of misidentifying orthologs.

| Identity % | BLOSUM62 | EX | delta-V | delta-P | # proteins | Length ave. |
|---|---|---|---|---|---|---|
| Protein length >1,102 amino acid with no gaps in alignment | | | | | | |
| >99 | 0.49 | 0.30 | 0.34 | 0.67 | 136 | 1532.7 |
| 98–99 | 0.44 | 0.25 | 0.32 | 0.63 | 137 | 1464.0 |
| 96–98 | 0.42 | 0.24 | 0.32 | 0.58 | 125 | 1539.4 |
| 87–96 | 0.38 | 0.20 | 0.31 | 0.55 | 57 | 1414.4 |
| Protein length >1,102 amino acid with gaps in alignment | | | | | | |
| >99 | 0.47 | 0.31 | 0.42 | 0.70 | 61 | 1659.3 |
| 98–99 | 0.41 | 0.24 | 0.34 | 0.63 | 119 | 1855.8 |
| 96–98 | 0.36 | 0.21 | 0.32 | 0.56 | 320 | 1792.7 |
| 87–96 | 0.33 | 0.19 | 0.31 | 0.50 | 437 | 1727.3 |
| Protein length 200–1,102 amino acid with no gaps in alignment | | | | | | |
| >95 | 0.43 | 0.25 | 0.32 | 0.63 | 6984 | 478.3 |
| 90–95 | 0.39 | 0.20 | 0.31 | 0.55 | 1229 | 407.9 |
| 80–90 | 0.38 | 0.20 | 0.32 | 0.52 | 276 | 350.9 |
| 60–80 | 0.41 | 0.25 | 0.37 | 0.57 | 91 | 372.8 |
| Protein length 200–1,102 amino acid with gaps in alignment | | | | | | |
| >95 | 0.38 | 0.22 | 0.32 | 0.59 | 2001 | 601.1 |
| 90–95 | 0.33 | 0.19 | 0.31 | 0.51 | 1529 | 566.4 |
| 80–90 | 0.30 | 0.17 | 0.30 | 0.47 | 1050 | 489.1 |
| 60–80 | 0.33 | 0.20 | 0.33 | 0.48 | 467 | 447.1 |

delta_P (*Braun, 2018*; *Pandey & Braun, 2020*). The results were largely similar. There was a general correlation between slower evolutionary rates and higher fractions of conservative changes, with a significant drop in the fraction of conservative changes between the slowest evolving, which was included in the slow set that has monkey-human identity >99% and protein length >1,102 amino acids, and the next slowest set (Figs. 1A and 1B). Proteins with alignment gaps showed similar or slightly lower fractions of conservative changes than those without gaps. We further studied the remaining proteins with shorter protein length (200–1,102 amino acids) divided into four groups (95–99%, 90–95%, 80–90%, and 60–80% identity), and found similar but less robust and consistent trends (Table 1 and Figs. 1C and 1D).

## Standing amino acid variants and evolutionary rates of proteins

We next studied the missense SNPs found in proteins with different evolutionary rates by using 1KG dataset (*Auton et al., 2015*). There were 15271 missense SNPs in the slowly evolving set of proteins (>1,102 aa with 99% identity and >304 aa with 100% identity) and 546297 missense SNPs in the fast set (all proteins that remain after excluding the slow
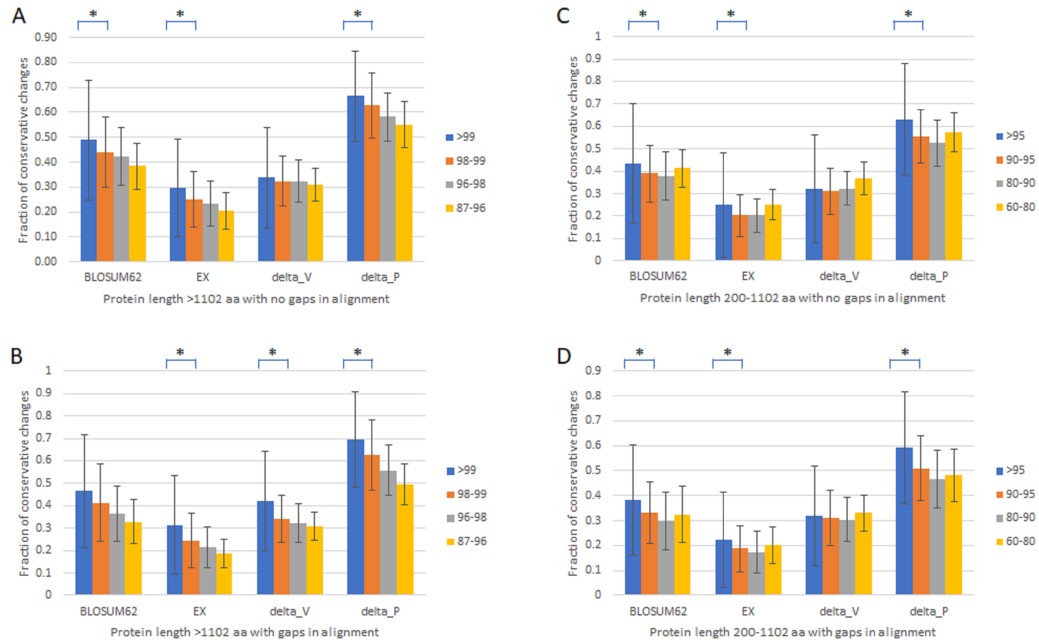

**Figure 1   Fraction of conservative substitutions in fixed changes in proteins of different evolutionary rates.** Shown are fractions of conservative changes in proteins of length >1,102 aa (A, B) or 200–1,102 aa (C, D) with either no gaps in alignment (A, C) and with gaps (B, D). *: $P < 0.05$. Chi-squared test.

set). We assigned each amino acid change found in a missense SNP a conservation score as described above. The number of SNPs in each score category was then enumerated. We performed this analysis by using each of the four different scoring matrixes and found largely similar results (Fig. 2). Missense SNPs in the slowly evolving set of proteins in general had lower fractions of drastic mutations, and higher fractions of conservative mutations relative to those in the faster evolving set of proteins (Fig. 2). The fraction of conservative mutations in the slow evolving set was significantly higher than that of the fast set ($P < 0.001$, Fig. 2).

To test for natural selection regarding conservative changes, we next divided the slowly evolving set of missense SNPs into three groups of different minor allele frequency (MAF) as measured in Africans (similar results were found for other racial groups). For fast evolving proteins at mutation saturation, low MAF values of a missense SNP would mean stronger negative selection, and so SNPs with low MAF are expected to have lower proportions of conservative amino acid changes, since these changes may mean too little functional alteration to be under natural selection. The results showed that for missense SNPs in the fast evolving set of proteins, the common SNPs with MAF >0.001 showed a higher fraction of conservative changes than the rare SNPs with MAF <0.001 ($P < 0.001$), indicating a stronger natural selection for the rare SNPs in the fast set (Fig. 3). While SNPs in the fast set showed similar fractions of conservative changes across three different MAF groups (>0.001, >0.01, and >0.05), there was a more obvious trend of having a higher proportion of conservative changes as MAF values increase from >0.001 to >0.01 to >0.05 for SNPs

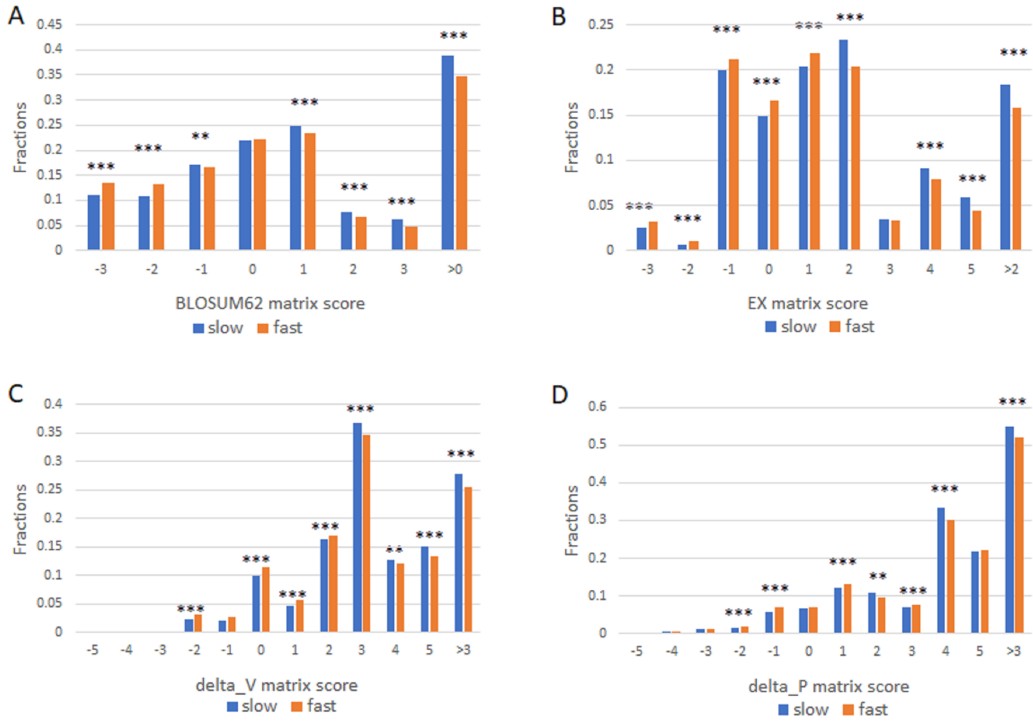

**Figure 2** **Fraction of conservative substitutions in standing missense substitutions in proteins of different evolutionary rates.** Missense SNPs from either the slow or the fast group of proteins were classified based on the scores in the four different matrixes as shown in A, B, C, and D. The fractions of each class are shown. \*\*\*, $P < 0.001$. \*\*, $P < 0.01$, Chi squared test.

in the slow set, consistent with weaker natural selection for common SNPs in the slow set (Fig. 3). Each of the three groups in the fast set showed a significantly lower fraction of conservative changes than the respective group in the slow set ($P < 0.01$), indicating stronger natural selection for SNPs in the fast set (Fig. 3). The results indicate that common SNPs in slowly evolving proteins had more conservative changes that were under a weaker natural selection.

## DISCUSSION

Our results here showed that fixed and standing changes in slowly evolving proteins were enriched with conservative amino acid substitutions. Similar results were obtained using four different matrixes to rank the conservative nature of a substitution. Based on substitution probability alone, amino acid substitutions in slowly evolving proteins are expected to be more conserved than those in fast evolving proteins, since fast evolving proteins have a higher probability of the doublet mutations that are necessary for a drastic substitution to occur, but have a very low rate of occurrence (*Whelan & Goldman, 2004*). If evolutionary time is not long enough for mutation saturation to occur, non-conservative substitutions would be expected to be a function of mutation rate and time. This simple explanation appears not to be the reason for the observations here, since the three matrixes

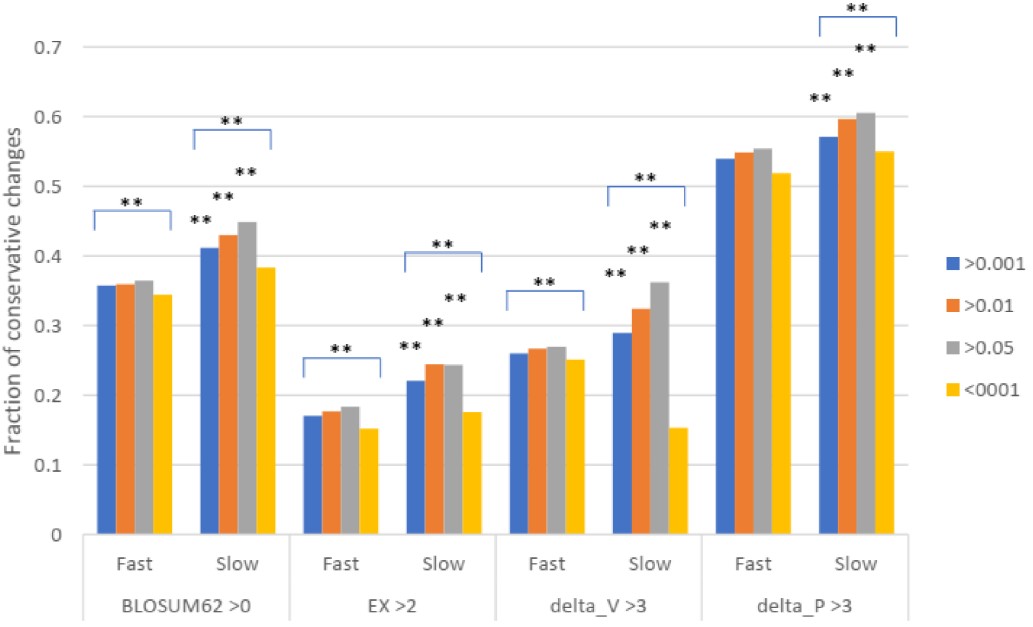

**Figure 3 Fraction of conservative substitutions in missense SNPs with different MAFs in proteins of different evolutionary rates.** SNPs from either fast or slowly evolving proteins were classified based on MAF values and the fractions of conservative changes in each class are shown. Statistical significance score in difference between slow and fast or between different MAF cutoffs are shown. **, $P < 0.01$, Chi squared test.

that have removed the impact of substitution probability produced similar results as the matrix that does not take into account the impact of substitution probability.

The four matrixes we used were developed in different ways: the BLOSUM62 matrix by using sequence alignments involving relatively divergent species, the delta_V and delta_P matrixes directly from the physicochemical properties of amino acids, and the EX matrix from experimental mutagenesis. However, the matrixes are still expected to provide information that applies to alignment data from relatively closely related species such as monkey and human, since proteins identified as fast evolving by comparing closely related species would also in general be identified as such by comparing more distantly related species. This is suggested by the molecular clock phenomenon or the constant evolutionary rates across time and species. It appears that the delta_V matrix produced less significant results compared to the other three matrixes. Overestimation of the number of conservative substitutions in fast evolving proteins may account for this. For example, substitutions involving differently charged residues with similar side chain volumes would be scored as non-conservative by the delta_P matrix but conservative by the delta_V matrix (e.g., Glu to Leu).

Our analysis does not take into account the co-evolution and co-variation of substitutions due to the physico-chemical constraints on protein structure and folding (*Pollock, Thiltgen & Goldstein, 2012*). Site specific variations in substitution constraints however may be similarly present in different proteins of different evolutionary rates so
that they may not affect the overall results here. Also, *Pollock, Thiltgen & Goldstein (2012)* show that site-specific preferences shift over time due to substitutions at other sites that are epistatic to the site of interest. Thus, it could be very complex to define site-specific preferences in a meaningful way.

It has recently been shown that coding region mutation rates as measured prior to the effect of natural selection are significantly lower in genes where mutations are more likely to be deleterious (*Monroe et al., 2020*). Mutations are more likely to be deleterious and less likely to be fixed in highly conserved proteins, which are by definition more common in slowly evolving proteins. Thus, slowly evolving genes in fact do have inherently slower mutation rates, which would make them less likely to reach mutation saturation.

The results here may be best accounted for by mutation saturation in fast evolving proteins, where multiple recurrent mutations at the same site have occurred across taxa and time (Fig. 4). At saturation, the range of mutations that have happened at any given site of any given taxon is irrelevant to the particular type of possible alleles the site may carry at present time. Natural selection is expected to play an important role in determining that. And natural selection of course would be most efficient if the mutated allele is functionally very different from the non-mutated allele. If two taxa are different in traits, it would follow that some of the differences in protein sequence between them would be non-neutral or non-conservative changes. Fast evolving genes play more adaptive roles and hence are more involved in accounting for the different traits, and so are expected to be enriched with non-conservative substitutions compared to slowly evolving genes. A fast evolving and adaptive site is more likely to be mutated more than once or encounter mutation saturation.

Fixed and standing conservative variants in slowly evolving proteins may be under weaker natural selection for several reasons. First, substitutions in slowly evolving proteins are more likely to be conservative and conservative changes may not alter protein structure and function as dramatically as the drastic changes, which may make it harder for natural selection to occur.

Second, as fixed variants cannot be fixed because of negative selection on the variants per se, they are either neutral or under positive selection. Indeed, fast evolving proteins are known to be under more positive selection (*Cai & Petrov, 2010*; *Yuan et al., 2017*), which implies that fixed variants in slowly evolving proteins can only become more neutral. Even if slightly deleterious mutations are fixed, it would not be because of selection but rather because of random drift. It makes sense for slowly evolving proteins to be spared by positive selection, because a mutation that takes a long time to arrive would be useless for quick adaptive needs.

Finally, SNPs in the slow set may be under negative selection if they produce drastic changes, or under no selection if they produce conservative changes (assuming no positive selection as explained above). While one would expect less conservative changes in the rare SNPs compared to the common SNPs, since negative selection may account in part for the low MAF value, the difference in the fraction of conservative changes between the rare SNPs and the common ones in the slow set should be greater than that in the fast set, since the SNPs in the fast set may be under natural selection regardless of MAF values (low

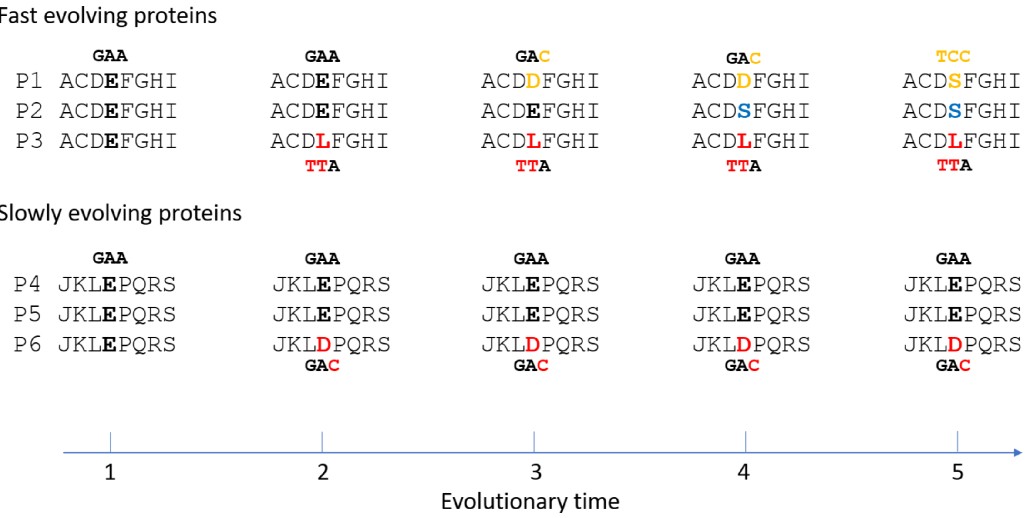

**Figure 4** **Illustration of non-conservative substitutions and mutation saturation in fast evolving proteins.** Three orthologous proteins from three different taxa are shown for either fast evolving (designated P1, P2, and P3) or slow evolving proteins (P4, P5, and P6). Nucleotide codon for residue E and its mutated codon in P1, P3, and P6 are also shown. A doublet mutation is found in the non-conservative substitution in P3 at time point 2, whereas only a single mutation is found in the conservative substitution in P6. Saturation phase for fast evolving proteins includes time point 3 to 5. The type of saturation we describe here is shown at time point 3 and 4 while the type seen in "long branch attraction" is shown at time point 5. At the saturation phase, for the fast evolving protein P1 to have the new allele D at time point 4 or S at time point 5 is largely a matter of natural selection.

MAF SNPs under more negative selection while high MAF SNPs under both positive and negative selection). Our results are consistent with such expectations.

If substitutions in fast evolving proteins are at saturation and under natural selection as indicated here, it would follow that genetic distances or degrees of sequence mismatches between taxa in these proteins would be at saturation, or no longer correlated exactly with time. It is easy to tell the difference between optimum/maximum saturation genetic distances and linear distances as described previously (*Huang, 2010*). Briefly, imagine a 100 amino acid protein with only 1 neutral site. In a multispecies alignment involving at least three taxa, if one finds only one of these taxa with a mutation at this neutral site while all other species have the same non-mutated residue, there is no saturation (Fig. 4, time point 2). However, if one finds that nearly every taxon has a unique amino acid, one would conclude mutation saturation as there would have been multiple independent substitution events among different species at the same site, and repeated mutations at the same site do not increase distance (Fig. 4, time point 3 and 4 for fast evolving proteins). We have termed those sites with repeated mutations "overlap" sites (*Huang, 2010*). So, a diagnostic criterion for saturated maximum distance between two species is the proportion of overlap sites among mismatched sites. Saturation would typically have 50–60% overlapped sites that are 2–3 fold higher than that expected before saturation (*Huang, 2010*; *Luo & Huang, 2016*). It is not expected to have near 100% overlapped sites, because certain sites may only accommodate 2 or very few amino acid residues at saturation equilibrium, which would
prevent them from presenting as overlapped sites even though they are in fact overlapped and saturated sites. Also, saturation may result in convergent evolution with independent mutations changing to the same amino acid (Fig. 4, time point 5 for fast evolving proteins). This overlap ratio method is an empirical one free of uncertain assumptions and hence more realistic than other methods of testing for saturation, such as comparing the observed number of mutations to the inferred one based on uncertain phylogenetic trees derived from maximum parsimony or maximum likelihood methods (*Philippe et al., 1994*; *Steel, Lockhart & Penny, 1993*; *Xia et al., 2003*).

By using the overlap ratio method, we have verified that the vast majority of proteins show maximum distances between any two deeply diverged taxa, and only a small proportion, the slowest evolving, are still at the linear phase of changes (*Huang, 2010*; *Luo & Huang, 2016*; *Yuan et al., 2017*). Variations at most genomic sites within human populations are also at optimum equilibrium, as evidenced by the observation that a slight increase above the present genetic diversity level in normal subjects is associated with patient populations suffering from complex diseases (*Gui, Lei & Huang, 2017*; *He et al., 2017*; *Lei & Huang, 2017*; *Lei et al., 2018*; *Yuan et al., 2012*; *Yuan et al., 2014*; *Zhu et al., 2015*), as well as the observation that the sharing of SNPs among different human groups is an evolutionary rate-dependent phenomenon, with more sharing in fast evolving sequences (*Yuan et al., 2017*). It is important to note that a protein in a complex species plays more roles than its orthologous protein in a species of less organismal complexity, as explained by the maximum genetic diversity hypothesis (*Hu et al., 2013*; *Huang, 2009*; *Huang, 2016*). A protein has more functions to play in complex organisms due in part to its involvement in more cell types, and hence it becomes more susceptible to mutational inactivation. While the divergence time among higher taxa such as between human and *Macaca* monkey is relatively short, mutation saturation could still happen for fast evolving proteins since the number of positions that can accept fixed substitutions is comparatively lower.

It is also important to note that the type of saturation we describe here is slightly different from that seen in "long branch attraction (LBA)" in phylogenetic trees (*Bergsten, 2005*). In LBA, saturation means convergent mutations leading to the same amino acid residue or nucleotide among (across) multiple taxa (Fig. 4, time point 5 for fast evolving protein P1 and P2). Although they were derived independently, these shared alleles can be misinterpreted in phylogenetic analyses as being shared due to common ancestry. However, for the type of saturation we have discussed here, independent mutations at the same site among different taxa would generally lead to different taxa having different amino acids rather than the same (Fig. 4, time point 3 and 4 for fast evolving proteins), since the probability of an independent mutation changing to the same amino acid is about 20 times lower than that of mutating to a different amino acid (assuming no difference in the probability of being mutated to among the 20 amino acids). Thus, the type of saturation we have described here is expected to be more commonplace in nature compared to that in the case of LBA. Since a single mutation is sufficient for a mismatch between any two taxa, multiple independent mutations at the same site leading to different amino acids would not increase the number of mismatches and would remain unnoticeable if one only aligns the sequences from two different taxa (Fig. 4, the number of mismatch between

P2 and P3 is 1 at time point 2 before saturation and remains as 1 at time point 4 after saturation). It only becomes apparent when one aligns the sequences from three different taxa (Fig. 4, time point 3 and 4 for fast evolving proteins), as we described above and in previous publications (*Huang, 2010*; *Luo & Huang, 2016*). However, even though the type of saturation we describe here does not increase the number of mismatches, it could result in a reduced number of mismatches in rare cases when independent mutations in two different taxa happen to lead to the same residue (Fig. 4, time point 5 for P1 and P2). Thus, it does not preclude the type of saturation observed in the case of LBA. These two types of saturation are essentially just two different aspects of the same saturation phenomenon, one more commonplace and manifesting as a higher overlap ratio while the other less common and manifesting as LBA.

It is well known that fast evolving proteins that have reached mutation saturation are not suitable for phylogenetic inferences. We have previously shown that mutation saturation as measured by the overlap ratio method has been largely overlooked (*Huang, 2012*; *Yuan et al., 2017*), in contrast to the long noted LBA. As mentioned above, it appears that the inherent mutation rates are different between fast and slowly evolving proteins as determined by studying the rate of fixed substitutions (*Monroe et al., 2020*). We can thus infer that if the rate difference is large enough, slowly evolving genes should be used in phylogenetic inferences because they would be less likely to reach mutation saturation. The findings here that fast evolving proteins are enriched with non-neutral substitutions relative to slowly evolving proteins are consistent with such an idea.

There are two points to note regarding fast and slowly evolving proteins. First, the definition of slowly evolving proteins here (99% identity) is only meant for the specific comparison between human and monkey. For relatively more distantly related species such as human and mouse, the set of slowly evolving proteins is expected to be similar but the percentage identity cutoff for the slow set would be lower than 99%. This is because proteins are known to evolve at constant rates across time and species according to the molecular clock and the neutral theory. Second, the classification of fast evolving proteins is not absolute and is evolutionary time-dependent. Proteins that are found as fast evolving or have reached mutation saturation after a certain relatively long time of evolution are expected to look like slowly evolving or not showing mutation saturation if evolutionary time is relatively short. This is supported by our results here of a nearly linear relationship between evolutionary rates and the fraction of non-conservative changes.

Our finding supports the possibility that, from early on since first diverging from a common ancestor, two sister species are expected to accumulate mostly neutral mismatches, which would later be replaced by non-conservative mismatches when time is long enough for mutation saturation to have taken place. This is to be expected as sister species should become more differentiated in phenotypes with time, and hence more different in sequences with time in terms of both the number of mismatches as well as the chemical nature (conservative or not) of the mismatches.

## CONCLUSION

Our study here addressed whether observed amino acids variants in slowly evolving proteins are more or less neutral than those in fast evolving proteins. The results suggest that fixed and standing missense variations in slowly evolving proteins are more likely to be neutral, and have implications for phylogenetic inferences.

## ACKNOWLEDGEMENTS

We thank the editor and reviewers for their valuable constructive advice and comments.

### Funding

This work was supported by the National Natural Science Foundation of China (81171880). The funders had no role in study design, data collection and analysis, decision to publish, or preparation of the manuscript.

### Grant Disclosures

The following grant information was disclosed by the authors:
National Natural Science Foundation of China: 81171880.

### Competing Interests

The authors declare there are no competing interests.

### Author Contributions

- Mingrui Wang conceived and designed the experiments, performed the experiments, analyzed the data, prepared figures and/or tables, authored or reviewed drafts of the paper, and approved the final draft.
- Dapeng Wang and Jun Yu performed the experiments, analyzed the data, authored or reviewed drafts of the paper, and approved the final draft.
- Shi Huang conceived and designed the experiments, analyzed the data, prepared figures and/or tables, authored or reviewed drafts of the paper, and approved the final draft.

### Data Availability

All the raw data were taken from public sources: The CCDS Project and Macaca mulatta:
https://www.ncbi.nlm.nih.gov/projects/CCDS/CcdsBrowse.cgi
https://jul2016.archive.ensembl.org/Macaca_mulatta/Info/Index.

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
