# Peer review of "Enrichment in conservative amino acid changes among fixed and standing missense variations in slowly evolving proteins"

_PeerJ, doi:10.7717/peerj.9983_

## Round 0.1 · original submission · Major Revisions

I would like to start by apologizing for the long time to decision. As you will see, the reviews are split with one very negative review. However, I see the potential in the manuscript so I wanted to give the authors the opportunity to revise the manuscript. Therefore, I wrote my own review to provide helpful guidance. Obviously, the authors should pay close attention to reviewer 1 and also address those comments.

Here is my editorial review:

I’ve read through the manuscript “Enrichment in conservative amino acid changes among fixed and standing missense variations in slow evolving proteins” (note that “slow evolving proteins” should read “slowly-evolving proteins”) and the two submitted reviews and I’d like to give some very explicit guidance regarding a potential revision. As you can see, the two reviewers are split, and you received one very negative review. Although I agree with the first reviewer that there are some major issues with the presentation, I also feel that the manuscript contains the core of a very good idea. For publication, I would like to see two things from the authors that constitute major revisions:

1) A reframing of the problem they seek to study.
2) Quantitative analyses showing that they are seeing a genuine enrichment of conservative changes in missense variants in slowly evolving proteins rather than a simple effect of the genetic code.

The reframing of the problem is, in a sense, relatively straightforward. The authors seem to have some major misconceptions regarding phylogenetics. Normally, I would have rejected a paper with such a large number of misconceptions regarding the current state of the art in phylogenetics, but I don’t think the problem they are studying is actually a phylogenetic problem per se. This is what convinced me to give the authors a chance to conduct a major revision.

The big misconception the author’s appear to have is related the state of the art regarding distance methods. It is simply untrue that “…the distance matrix methods are sound provided that one uses neutral variants that accumulate to increase genetic distances in a nearly linear fashion common to the species concerned.” (their lines 68-70). Their statement (lines 259-260) that “Distance matrix methods do not rely on such models but requires the molecular clock and hence the neutral variants.” This statement regarding the molecular clock is true for some distance methods, such as UPGMA. However, as Sanderson and Kim (2000) stated “...performance evaluations [of phylogenetic methods] based on computer simulations (reviewed in Hillis et al., 1994; Li, 1997) and studies of ‘known phylogenies’ (Russo et al., 1996; Naylor and Brown, 1998; Leitner and Fitch, 1999). However, little consensus has emerged, except that a few methods that are not widely used anyway, such as UPGMA, perform poorly” (see the original for the citations embedded in the quote). In other words, the distance methods that require a molecular clock are known to perform poorly have been known to perform poorly for at least two decades and are therefore seldom, if ever, used.

The reality is that most commonly used distance methods, such as neighbor-joining (Saitou and Nei 1987), BioNJ (Gascuel 1997), and minimum evolution (Rzhetsky and Nei 1993; Desper and Gascuel 2002). Indeed, least squares methods, like the Fitch-Margoliash method that the authors cite in their manucript, can be used for taxa without a clock unless one imposes the additional requirement that the least squares trees be ultrametric. In other words, Fitch-Margoliash can be used with data that are not clock-like unless one imposes the assumption of a clock (this is the difference between the fitch and kitch methods in Phylip – the former does not assume a clock whereas the latter does).

Likewise, the criticism of maximum likelihood and Bayesian methods implicit in their statement that among “…the existing methods of phylogeny inferences, most, such as the maximum likelihood methods and Bayesian methods, require the assumption of certain evolutionary models of amino acid or nucleotide changes, which may be unrealistic (Felsenstein 1981; Rannala, Yang 1996)” (lines 256-258) is unfounded. It is true that proofs of consistency for ML phylogenetic estimation require the generating model to be correct. However, there are models of evolution that incorporate positive selection (e.g., Yang and Nielsen 2000; although I would acknowledge that those models are seldom used for tree estimation; they are primarily used to study the process of protein evolution given “known” trees). But that issue is largely beside the point; the GTR model and its sub-models essentially assume neutrality and they appear to be fairly robust under most circumstance; the problem cases are limited to very short branches. Indeed, this is the whole basis for the field of phylogenetics – if ML and Bayesian methods were so dependent on the minutia of model fit that they fundamentally don’t work unless the model is perfect (a criticism implicit in their lines 256-258) the members of the phylogenetic community would have noticed by now!

In other words, I feel the business of justifying their work with reference to the molecular clock (or phylogenetics in general) is somewhat misguided. The interaction between clock-like evolution and the molecular clock is complex. Clock-like evolution is certainly helpful and can allow simple models of evolution (whether they are used in a maximum likelihood, Bayesian, or even a distance framework) to perform better than might be naively expected (Bruno and Halpern 1999). But clock-like evolution is not a prerequisite for any commonly used method of phylogenetic estimation.

I’ve spent a lot of time criticizing the authors’ framing of the problem with their reference to the field of phylogenetics, but their study is really a molecular evolution study and not a phylogenetic study. This, along with the second reviewer’s relatively positive evaluation of the manuscript, is what encouraged me to offer “major revisions” rather than “reject.”

There is one relatively large (and potentially problematic) issue with the author’s analysis. On average, amino acid substitutions that require a single substitution tend to be more conservative than those requiring more than one substitution. Thus, their results might reflect something intrinsic to the structure of the genetic code. I think the authors could deal with this by using another way to assess conservative vs. non-conservative amino acid changes. The author’s use of the log-odds scores from the BLOSUM62 matrix to establish conservative vs. non-conservative substitutions suffers from the fundamental problem that the effects of the genetic code are buried in the BLOSUM62 matrix. In other words, the log-odds of amino acid i being aligned with amino acid j reflects two things: the probability of a missense mutation leading to an i to j polymorphism and the probability that j will be fixed. You are (de facto) using the BLOSUM62 matrix as a proxy for the fixation probability, which you assume to reflect the “conservativeness” of the amino acid exchange. However, the strong evidence that the “instantaneous” doublet nucleotide substitution rate is very low (Whelan and Goldman 2004) means that the first issue the probability of a missense mutation leading to an i to j polymorphism should have a big impact on the values in the BLOSUM62 matrix (or any empirical evolutionary matrix, including the original Dayhoff et al. [1978] PAM matrices or the Gonnett et al. [1992] matrix).

There is a simple solution to this problem. There are matrices of conservativeness that remove the impact of substitution probability. Specifically, there is the “EX” matrix of Yampolsky and Stoltzfus (2005), which is based on laboratory mutagenesis. There are also physicochemical matrices in Braun (2018); I would recommend using EX, delta-V (normalized change in amino acid side chain volume), and delta-P (normalized change in amino acid side chain volume). All three of these changes can be obtained from matrices in spreadsheets that are available from github (https://github.com/ebraun68/clade_specific_prot_models; this is the github site for Pandey and Braun 2020). Specifically, the EX matrix (or, more accurately, a normalized symmetric version of the EX matrix) is in the excel spreadsheet EX_matrix_sym.xlsx; the delta-V and delta-P matrices can be found in one of the sheets (the sheet called “Exchanges”) in the file exchange_Pandey_Braun.xlsx.

I think that reframing the manuscript as simply an exploration of molecular evolution and using some matrices that are unaffected by the structure of the genetic code (i.e., the EX, delta-V, and delta-P matrices) will greatly improve the manuscript. On lines 113-115 the authors state that the “…degree of physical/chemical change in an amino acid missense mutation was ranked by a scoring series, -3, -2, -1, 0, 1, 2, 3, in the BLOSUM62 matrix with more positive values representing more conservative changes.” It seems to me that they could do something very similar with the three matrices I recommend. The only major difference is the range of values (all three of the matrices I recommend are normalized to range from zero to one) of the fact that the matrices include non-integer values. However, the same idea applies – values closest to one are more conservative and values closer to zero are less. If the authors wanted to work with integers (perhaps this would be easier with their code) they could generate integer versions of the matrices by multiplying by constant, subtracting half of that constant, and rounding to the nearest integer (e.g., if they multiply by 10, subtract 5, and round they would get a matrix of values that range from -5 to +5 with higher positive values reflecting more conservative changes. I feel this approach would overcome the fundamental flaw of using BLOSUM62 for this analysis.

It would certainly be fine to keep the BLOSUM62 analyses you have already conducted. Likewise, it would be fine to acknowledge that your results might imply that distance analyses of slowly evolving loci are better. There is actually a growing literature that distance methods may have desirable properties not necessarily related to the clock but rather to the multispecies coalescent (Dasarathy et al. 2015; Rusinko and McPartlon 2017; Allman et al. 2019). Thus, empirical hints for working with distance methods might be valuable. Of course, there are other reasons to avoid rapidly evolving loci (i.e., avoiding saturation). My big point is that this manuscript should not primarily focus on phylogeny and instead focus on molecular evolution.

I would like to say three addition things. First, I think should endeavor to improve the English language presentation of their work. I am always a bit worried about saying this because the need for good English writing puts an additional burden on researchers who are not native English speakers. But (perhaps unfortunately) it is simply a fact that the international language of science at this point is English. Second, I thought long and hard about whether the manuscript would merit publication if using the EX, delta-V, delta-P matrices showed that their core finding (that slowly-evolving proteins are enriched for conservative amino acid changes relative to rapidly evolving proteins) is incorrect (or more accurately, that the conclusion only emerges when the BLOSUM62 matrix is used to score conservative vs. non-conservative changes). I think it would be interesting regardless, as long as the results are sound. Finally, I recognize that I am asking the authors to use information from two of my papers (Braun 2018 and the Pandey and Braun 2020 bioRxiv preprint). I don’t like to push citations of my own work too strongly when I am editing papers, but in this case it is only two papers and I feel that specific work is directly relevant and – in the case of the EX, delta-V, and delta-P matrices – directly addresses what I believe to be a fundamental limitation of using the BLOSUM62 matrix to score conservative vs non-conservative changes.

I hope this guidance is helpful. I do think this paper has potential, but I also believe these improvements are essential to make the manuscript publishable. Note that I do plan send the manuscript out for re-review if it is resubmitted in a revised version.
* * *
References for this editorial review:

Allman, E. S., Long, C., & Rhodes, J. A. (2019). Species tree inference from genomic sequences using the log-det distance. SIAM Journal on Applied Algebra and Geometry, 3(1), 107-127.

Braun, E.L. (2018) An evolutionary model motivated by physicochemical properties of amino acids reveals variation among proteins. Bioinformatics, 34, i350-i356.

Bruno, W. J., & Halpern, A. L. (1999). Topological bias and inconsistency of maximum likelihood using wrong models. Molecular Biology and Evolution, 16(4), 564-566.

Dayhoff, M.O. et al. (1978) A model of evolutionary change in proteins. In Dayhoff, M.O. (ed.), Atlas of Protein Sequence and Structure, National Biomedical Research Foundation, Silver Springs, MD, Vol. 5, pp. 345-352.

Desper R, and Gascuel O. Fast and accurate phylogeny reconstruction algorithms based on the minimum-evolution principle J Comput Biol., 2002, vol. 9 (pg. 687-705)

Dasarathy G., Nowak R., & Roch S. Data requirement for phylogenetic inference from multiple loci: a new distance method. IEEE/ACM Trans. Comput. Biol. Bioinforma, 12 (2) (2015), pp. 422-432

Gascuel, O. (1997). BIONJ: an improved version of the NJ algorithm based on a simple model of sequence data. Molecular biology and evolution, 14(7), 685-695.

Gonnet, G. H., Cohen, M. A., & Benner, S. A. (1992). Exhaustive matching of the entire protein sequence database. Science, 256(5062), 1443-1445.

Pandey, A. & Braun, E.L. Protein evolution is structure dependent and non-homogeneous across the tree of life. bioRxiv 2020.01.28.923458; doi: https://doi.org/10.1101/2020.01.28.923458

Rzhetsky, A., & Nei, M. (1993). Theoretical foundation of the minimum-evolution method of phylogenetic inference. Molecular biology and evolution, 10(5), 1073-1095.

Rusinko, J., & McPartlon, M. (2017). Species tree estimation using Neighbor Joining. Journal of theoretical biology, 414, 5-7.

Saitou, N., & Nei, M. (1987). The neighbor-joining method: a new method for reconstructing phylogenetic trees. Molecular biology and evolution, 4(4), 406-425.

Sanderson, M. J., & Kim, J. (2000). Parametric phylogenetics? Systematic Biology, 49(4), 817-829.

Whelan, S., & Goldman, N. (2004). Estimating the frequency of events that cause multiple-nucleotide changes. Genetics, 167(4), 2027-2043.

Yampolsky, L.Y. and Stoltzfus, A. (2005) The exchangeability of amino acids in proteins. Genetics, 170, 1459-1472.

Yang, Z., & Nielsen, R. (2000). Estimating synonymous and nonsynonymous substitution rates under realistic evolutionary models. Molecular biology and evolution, 17(1), 32-43.

Reviewer 1 ·

Basic reporting

no comment

Experimental design

no comment

Validity of the findings

The primary finding that amino-acid substitutions and polymorphisms in slowly evolving (conserved) genes are more conservative than those in non-slow genes is sensible and is documented reasonably by the authors.

However, I have a concern about the authors use of the term "saturation" and their interpretation of the results.

Based on the Discussion, the authors seem to take a somewhat idiosyncratic view of what "saturation" means. For a sequence to be saturated with substitutions I think generally implies that so many repeated substitutions occurred along a particularly long branch(es) that the stationary distribution has more or less been approached in subsets of the data, which can cause artificial attraction among the longest branches when inferring phylogenetic trees ("long branch attraction"). It is hard to imagine such problems occurring in human-macaque comparisons, since these are two very closely related species by any standard. The authors instead seem to be discussing a problem where individual sites are saturated with substitutions. However, their definition seems to be that any site with two or more apparent substitutions is "saturated". If this is really what they are arguing, it does not seem correct. Even parsimony based methods can deal with such cases. Perhaps they are interested in some pathology of distance-based methods, but I can't quite figure out what they are actually claiming. For the most part, I think this problem affects only the few places throughout the manuscript where "saturation" is mentioned. However, the second half of the discussion seems heavily based on the idea, which I feel is insufficiently justified and incompletely explained. Perhaps the authors can clarify why their definition of saturation is so different from the standard view from statistical phylogenetics. I recognize this aspect of the study seems to be connected with other work by these authors, however, I feel that the discussion of these matters is somewhat disconnected from the study, which is otherwise rather unobjectionable. I therefore suggest that for publication either a careful exposition of these ideas be presented that tries to reconcile the standard definition of saturation with theirs, or that this part of the discussion be removed.

One other spot where the results seem over interpreted is where the authors state "our findings here are consistent with the view that saturation is maintained by positive selection as fast evolving proteins have lower fraction of conservative changes, and inconsistent with the presently popular view that variant sites at saturation are fully neutral." First, fast evolving proteins include both neutrally evolving and positively selected genes. Second, this hardly seems like any kind of evidence that "saturation is maintained by positive selection." Third, I cannot see how any one could think that "variant sites at saturation are fully neutral". These ideas need to be more clearly expressed and justified. However, more importantly, they do not seem to me to follow from the results of the paper.

Reviewer 2 ·

Basic reporting

English used throughout the manuscript is poor and this single factor has great impact on the value of the whole study. For example, the first sentence of the whole manuscript "Proteins were first used in the early 1960s to discover the molecular clock" is inaccurate. It should be "Protein sequences were first used...". There are many places throughout the manuscript contain this type of inaccuracy and irregular use of common English. For instances, Line 68 "methods are sound provided that..." ?? and Line 74 "would cease..."

Experimental design

Authors should provide a toy example of sequences to illustrate their main point. The whole bioinformatics experiment design and procedure look puzzling.

Validity of the findings

From the nonstandard use of terminology, it seems authors who wrote the manuscript are not familiar with the fields, molecular evolution and pouplation genetics. For example, line 52-53 "gene non-identity between species" should be "sequence divergence between species". Line 76 "negative selection as defined by dN/dS ratio" should be "negative selection as measured using dN/dS ratio". Line 78-80 is difficult to understand what authors are talking about.

Additional comments

Please define "conservative mismatches".

---

## Round 0.2 · Minor Revisions

I would like to apologize for the delay in this response. I had difficulty securing additional reviews. Fortunately, I was able to secure a review that I consider very thorough so I have decided to move forward rather than waiting any longer on the second review.

I provided a fairly long editorial review for the initial submission and I feel that you addressed it quite well. I had some concerns in terms of the framing of the question, which you addressed quite well. My other concern, which was much more fundamental, was that your use of the BLOSUM matrix introduces circularity. You addressed that concern quite well.

I think you should read the new reviewer's comments thoroughly and do your best to address them. I would like to offer the following guidance for a minor revision:

First, the reviewer makes a good point that all of the matrices (BLOSUM, delta V, delta P, and EX) are crude measures in the sense that they do not consider any site-specific variation. I think the reviewer's concerns are extremely valid, but I think you can address those concerns with a verbal argument rather than conducting any new analyses.

I've spent a fair amount of time thinking about whether there is any easy way to capture among-sites variation in selection on sites (i.e., differentiating sites that tolerate only a specific subset of amino acids from those sites that tolerate a broader set of amino acids). I think it would be extremely difficult. One approach would be to use proteins structure, which would fall outside of the scope of the manuscript. You could use some sort of PSSM approach, as the reviewer suggests, but that would require mapping information from the PSSM onto the sequences. Although alignments that could be converted into PSSMs in this manner do exist (i.e., Pfam) but using them seems like a fundamentally different study.

The Pollock et al. paper (2012; PMID: 22547823) that the reviewer cites actually makes a point that could be useful for a verbal argument. Specifically, Pollock et al. show that site-specific preferences shift over time due to substitutions at other sites that are epistatic to the site of interest. The point I would make about this is simply that it could be very complex to define site-specific preferences in a meaningful way. The fact that you see a significant difference between “slow” and “fast” proteins when you use the crude matrices suggests to me that you are onto something important. It is possible that undertaking an effort to define site-specific preferences using PSSMs or structure would refine things further (e.g., maybe delta P has a strong impact in some sites and delta V has an impact in others) but I think this is a different study. What you have is interesting as is. You likely have other thoughts on the topic as well; my big suggestion is to provide a brief discussion of the fact that different sites in proteins tolerate a specific subset of amino acids, that the processes underlying this are complex, and that this fact is unlikely to invalidate your fundamental conclusion.

Second, the reviewer also asked for a little more speculation about the implications of your findings for phylogenetic analyses. I would recommend keeping any speculation relatively brief. It does seem to me that your results imply evolutionary rate should have an impact on the best-fitting model. This is not surprising and could be stated in a simple manner. Obviously, the best-fitting model has implications for maximum likelihood, Bayesian, and even for some distance analyses (e.g., distance analyses that use ML estimates of distances) in phylogenetics.

Finally, the reviewer’s question about why ~25% of proteins encoded by the macaque and human genomes are not considered is a good one. I assumed in the previous submission that you used all orthologous pairs you could, omitting cases where orthology was unclear or one of the sequences was problematic in some way. It would be nice for you to address this issue in a direct manner.

Overall, I would recommend a fairly focused revision, trying to address comments relatively narrowly. Of course, a very thorough examination of the manuscript for typos (the reviewer points out two minor errors) is very appropriate and I hope you will take the time to do this.

I hope this was helpful and I look forward to seeing a revision.

Reviewer 3 ·

Basic reporting

This revised manuscript is certainly an improved version as far as the focus and description of the research is concerned. The manuscript describes interesting findings and is clear and understandable. But it needs more elaboration and context of the analysis presented; in the introduction, and in other sections as well. For instance, Line 72 “We here found that the proportion of conservative substitutions between species….”- it would help if the introduction would include a description of what type of ‘conservative’ substitutions are being referred to. The description does not appear until the results section. In addition, it would help to specify that the analysis is limited to genome sequences of two [very] closely related species.

Likewise, the scientific names of the species being studied are not mentioned until later in the manuscript.

In the results/discussion there should be a description, even if brief, as to, (a) why results based on the delta_V matrix are different from other substitution matrices, (b) the context in which these substitution matrices were derived, compared to the BLOSUM matrices, i.e. estimates of substitution rates from sequences in relatively closely related clades. Similarly, the tertiary structural context of the estimated substitutions, which is relevant to the discussion of the strength of selection on the measured mutations would be appropriate.

Experimental design

Methods used to identify and quantify conservative mutations are adequate, even if simplistic, for the patterns that the authors seek to characterize. But the description can be more explicit. For example, since all the analyses are based on pairwise sequence alignments (one sequence per species) generated by the BLASTP algorithm, it will be useful to know if the alignments were generated using the BLOSUM-62 matrix (the default for the BLASTP algorithm), even when the other substitution matrices (EX, delta_V and delta_P) were used to identify and quantify the conservative substitutions in the sequence alignments. It is relevant to know if there are any differences in the alignments generated using the BLOSUM-62 matrix compared to alignments generated using the other matrices, and if the variation in alignment would affect the quantification of conservative substitutions. This could be interesting, if not a cause for concern, for fast-evolving proteins (as defined by the authors) that have lower sequence identities, especially for gapped alignments.

Performance of BLASTP alignments can often be weak when comparing sequences that show medium to high divergence (but not necessarily sequences derived from highly divergent species). See also comment regarding Table 1.

Validity of the findings

The main findings that conservative substitutions are enriched in slowly evolving proteins, and that it is correlated with minor allele frequency is interesting. The conclusions are reasonable in the context of the measurements reported. However, there are some caveats given, (a) the very strict definition/classification of the slowly evolving proteins (99% identity), (b) that the analyses are limited to pair-wise sequence alignments, and (c) restricted to very closely related species. By contrast, how different are the results likely to be if we were to extend (or repeat) the analyses using multiple species/alignments instead of pairwise-alignments? The authors’ definition of slowly evolving proteins (99% identity) may not be broadly applicable. It is quite likely that the results will be different even if proteins from multiple species of the genus Macaca were to be analyzed, let alone comparing many species from different genera, or larger clades such as Old World and New World monkeys, or primates as a whole. The point is, the limitations of the pairwise comparisons should be discussed, rather than the broad implications that the authors seem to indicate. Presumably the fraction of proteins that can be classified as ‘slowly evolving’ might drop substantially. Ideally, running the analysis on multiple alignments of a few orthologous families, if not for all the proteins pairs, would be useful (see also a related note regarding Table-1 under “Comments for the author”).

This is not to question the validity of the findings/conclusions, but to consider the limitations of the experimental design and the context of the measurements, as well as the extent to which the findings can be extrapolated. The article (and readers) would benefit in general if issues of more direct relevance to the core results are discussed, in addition to the explanation of what saturation means. For example, why the trends are different in quantifications based on the delta_V matrix compared to other substitution matrices in Figure 1.

It would be valuable to also discuss why/where pairwise comparisons would be relevant and useful, and where they may not be. After all, the toy example in figure 4 is more relevant to multiple alignments for comparing multiple taxa. Saturation in faster evolving proteins is not inconceivable given the estimated divergence times of several to many millions of years between closely related species.

In addition, although the EX, delta_P and delta_V matrices remove the probabilistic element compared to the BLOSUM matrices, they are coarse-grained estimates. These matrices do not consider site-specific substitution patterns, as in PSI-BLAST and the PSSM used therein, which is relevant to multiple sequence alignments and multi-species comparisons. This may be relevant to conclusions about the strength of selection. For instance, the relatively simplistic linear sequence-based pairwise measurements used by the authors do not take into account the co-evolution and co-variation of substitutions due to the physico-chemical constraints on protein structure and folding (see Pollock et al, 2012, PMID: 22547823). In this context, the suitability and efficacy of the dN/dS ratios to measure selection has been criticized (see Drummond and Wilke, 2008, PMID: 18662548). The EX, delta_P and delta_V matrices (Braun, PMID: 29950007; Braun and Pandey, 2020, doi: 10.1101/2020.01.28.923458) that explicitly consider the effects/context of protein structure, in addition to the population parameters. The delta_P and delta_V matrices were derived from soluble proteins; transmembrane proteins were excluded from the estimations (see also next paragraph). Therefore, a discussion of the coevolution of sequence in the context of protein structure, and the efficacy of dN/dS for measuring the extent of selection are directly relevant to the analyses and results presented.

Going back to the performance of BLASTP alignments that was mentioned in the Experimental Design section, the data presented in Table 1 indicates a potential issue. This could also indicate a limitation of the current experimental design. Assuming that the numbers of proteins in the different categories (column 6) are non-overlapping sets of proteins, I counted a total of 15,019 proteins (or protein pairs?). Given that both human and Macaca genomes are estimated to have about 20,000 protein coding genes, I am curious why roughly a fourth of the proteins are not included in the analyses. Do they produce poor alignments?

Regarding implications for phylogenetic inference, a discussion, even if brief, would be valuable instead of simply stating “…. and have implications for phylogenetic inferences” (line 319). This brief prelude is more appropriate in the abstract.
Since phylogenetic analysis is always based on multiple alignments, how would the findings here be applicable. What are the implications? What type of phylogenetic inference: Of very closely related species of the Macaca genus? Or of primates in general? Or more general?

Additional comments

Minor edits/revisions:

Table 1: It will be useful to give an overview of the fraction of proteins per species analyzed: either a pie chart or a Venn diagram or both.

Line 77: "... less natural selection" should be “weaker natural selection”?
Line 225: Fixed and "stranding" should be "standing"?

Please check for similar issues.

---

## Round 0.3 · Minor Revisions

I am really excited by this manuscript and am happy to accept it, but there are three very minor corrections I'd like you to make before the final acceptance. One is a factual matter, one is a minor grammatical issue, and the last is a matter of consistency. I will provide the exact locations and some suggested fixes:

The first is the paragraph on lines 220-230 (note that I am using line numbers from the pdf; line numbers in word documents with tracked changes will differ). I suggest you use:

The four matrixes we used were developed in different ways: the BLOSUM matrix by using sequence alignments involving relatively divergent species, the delta_V and delta_P matrixes directly from the physicochemical properties of amino acids, and the EX matrix from experimental mutagenesis. However, the matrixes are still expected to provide information that applies to alignment data from relatively closely related species such as monkey and human, since proteins identified as fast evolving by comparing closely related species would also in general be identified as such by comparing more distantly related species. This is suggested by the molecular clock phenomenon or the constant evolutionary rates across time and species. It appears that the delta_V matrix produced less significant results compared to the other three matrixes. Overestimation of the number of conservative substitutions in fast evolving proteins may account for this. For example, substitutions involving differently charged residues with similar side chain volumes would be scored as non-conservative by the delta_P matrix but conservative by the delta_V matrix (e.g., Glu to Leu).

The changed material is this:

"The four matrixes we used were developed in different ways: the BLOSUM matrix by using sequence alignments involving relatively divergent species, the delta_V and delta_P matrixes directly from the physicochemical properties of amino acids, and the EX matrix from experimental mutagenesis. However, the matrixes are still expected to provide information that applies to alignment data ..." (this is the beginning of the paragraph)

This originally stated that all for matrixes were based on alignments of divergent taxa. This is not correct (although I think I see where in the reviews you may have taken this comment from). Two matrixes were based solely on chemistry and a third was based on laboratory mutagenesis studies.

The second issue is the paragraph on lines 238-243, which should be changed as follows:

It has recently been shown that coding region mutation rates as measured prior to the effect of natural selection are significantly lower in genes where mutations are more likely to be deleterious (Monroe et al. 2020). Mutations are more likely to be deleterious and less likely to be fixed in highly conserved proteins, which are by definition more common in slowly evolving proteins. Thus, slowly evolving genes in fact do have inherently slower mutation rates, which would make them less likely to reach mutation saturation.

The changes are minor - something was missing in the next to last sentence so I added "more common in". I also added a comma after the "Thus" that begins the last sentence.

The final thing I'd like you to do is check the names you use for the delta matrixes. You sometimes use "delta P" and "delta V", other times you use "delta-P" and "delta-V" and other times you use "delta_P" and "delta_V". I don't have a strong feeling regarding the best way to write this out - I would be happy with either. However, you should pick one of the ways of writing the matrix names and stick with it throughout the paper. I note that you use "delta P" and "delta V" in the figures, so perhaps it is best to use that (although I will leave the final decision in your hands).

I apologize for feeling that I have to ask for these final changes rather than a simple accept since these are minor issues. But I'd really like to see this manuscript get out the door and into the hands of readers, who will hopefully be as interested in your results as I am. I hope it will be easy for you to get this done quickly so I can also be quick in the final acceptance. Best wishes!

---

## Round 0.4 · accepted · Accept

Thank you so much for making the few final corrections. It has been a delight to edit this manuscript and I'm excited to see this. As you know, there were a few minor changes that I believed to be necessary before publication and I thank you for making those edits.

I apologize that I have been slower than normal with editing this manuscript. I hope you understand how disrupted things have been here; I've been trying my best. I'm just glad that I was able to help a little bit with your excellent manuscript!